# Income inequality and mental health in adolescents during COVID-19, results from COMPASS 2018–2021

Claire Benny[1,2]*, Ambikaipakan Senthilselvan[1], Karen A. Patte[3], Brendan T. Smith[2,4], Paul J. Veugelers[1], Scott T. Leatherdale[5], Roman Pabayo[1]

1 Edmonton Clinic Health Academy, University of Alberta School of Public Health, Edmonton, Alberta, Canada, 2 Public Health Ontario, Toronto, Ontario, Canada, 3 Department of Health Sciences, Brock University, St. Catharines, Ontario, Canada, 4 Dalla Lana School of Public Health, University of Toronto, Toronto, Ontario, Canada, 5 School of Public Health Sciences, University of Waterloo, Waterloo, Ontario, Canada

* cbenny@ualberta.ca

## Abstract

### Introduction

Understanding the inequitable impacts of the ongoing COVID-19 pandemic on youth mental health are leading priorities. Existing research has linked income inequality in schools to adolescent depression, however, it is unclear if the onset of the pandemic exacerbated the effects of income inequality on adolescent mental health. The current study aimed to quantify the association between income inequality and adolescent mental health during COVID-19.

### Material and methods

Longitudinal data were taken from three waves (2018/19 to 2020/21) of the Cannabis, Obesity, Mental health, Physical activity, Alcohol, Smoking, and Sedentary behaviour (COMPASS) school-based study. Latent Growth Curve modelling was used to assess the association between Census District (CD)-level income inequality and depressive symptoms before and after the onset of COVID-19.

### Results

The study sample included 29,722 students across 43 Census divisions in British Columbia, Alberta, Ontario, and Quebec. The average age of the sample at baseline was 14.9 years [standard deviation (SD) = 1.5] and ranged between 12 and 19 years of age. Most of the sample self-reported as white (76.3%) and female (54.4%). Students who completed the COMPASS survey after the onset of COVID reported 0.20-unit higher depressive scores (95% CI = 0.16, 0.24) compared to pre-COVID. The adjusted analyses indicated that the association between income inequality on anxiety scores was strengthened following the onset of COVID-19 ($\beta$ = 0.02, 95% CI = 0.0004, 0.03), indicating that income inequality was associated with a greater increase in anxiety scores during COVID-19.

(contact via sherry.rezvani@uwaterloo.ca) for researchers who meet the criteria for access to confidential data.

**Funding:** The COMPASS study has been supported by a bridge grant from the CIHR Institute of Nutrition, Metabolism and Diabetes (INMD) through the "Obesity – Interventions to Prevent or Treat" priority funding awards (OOP-110788; awarded to SL), an operating grant from the CIHR Institute of Population and Public Health (IPPH) (MOP-114875; awarded to SL), a CIHR project grant (PJT-148562; awarded to SL), a CIHR bridge grant (PJT-149092; awarded to KP/SL), a CIHR project grant (PJT-159693; awarded to KP), and by a research funding arrangement with Health Canada (#1617-HQ-000012; contract awarded to SL). The COMPASS-Quebec project additionally benefits from funding from the Ministère de la Santé et dttmes Services sociaux of the province of Québec and the Direction régionale de santé publique du CIUSSS de la Capitale-Nationale. The COMPASS study is also funded by a SickKids Foundation New Investigator Grant, in partnership with CIHR Institute of Human Development, Child and Youth Health (IHDCYH) (Grant No. NI21-1193; awarded to KP) funds a mixed methods study examining the impact of the COVID-19 pandemic on youth mental health, leveraging COMPASS study data. The current study has been supported by a Women and Children's Health Research Institute grant (#3161; awarded to RP) and a CIHR Operating Grant: Data Analysis Using Existing Databases and Cohorts (awarded to RP). RP is a Tier II Canada Research Chair in Social and Health Inequities Across the Lifespan. No sponsors of the investigators nor the COMPASS study have played any role in the study design, data collection and analysis, decision to publish, or preparation of the manuscript.

**Competing interests:** The authors have declared that no competing interests exist.

## Discussion

The adjusted results indicate that the association between income inequality and adolescent anxiety persisted and was heightened at the onset of COVID-19. Future studies should use quasi-experimental methods to strengthen this finding. The current study can inform policy and program discussions regarding the effects of the COVID-19 pandemic and pandemic recovery for young Canadians and relevant social policies for improving adolescent mental health.

## 1.0 Introduction

The coronavirus 2019 (i.e., COVID-19) pandemic has adversely affected population mental health worldwide [1], including the mental health of Canadian youth. Youth reported the largest reductions in mental health following the COVID-19 pandemic and restrictive measures, compared to any other age group in Canada, with 41.5% indicating they had excellent or very good perceived mental health in 2020, compared to 62.1% pre-pandemic in 2018 [2]. Further, 57% of adolescents report that their mental health has worsened since the onset of school closures and "lockdowns" [2]. School closures and lockdowns were likely associated with feelings of loneliness or social isolation, which can contribute to adolescents' risk of poor mental health [3]. While, initially, early public health measures in Ontario reduced mental health out-patient visits in adolescents drastically in March/April of 2020 due to lockdowns and a shift to online services, usage of physician-based mental health services rose to above pre-COVID rates as of July 2020 and remained at this level until at least February 2021 [4]. Emergency department visits among adolescents for mental health-related services also increased in Canada following the pandemic by as much as 29.7% [5].

Data on the effects of the COVID-19 pandemic on income inequality are currently lacking, given the recency of the pandemic and the infrequency of income inequality measures. It is plausible that the pandemic widened the gap between the highest and lowest earners (i.e., income inequality) in Canada, with increases in unemployment rates disproportionately affecting the lowest earners, those less educated, immigrants, people of colour, and those with precarious employment [6]; that said, income inequality may have also been subsequently decreasing owing to the Canadian Emergency Response Benefit (CERB) and the Canadian Emergency Student Benefit (CESB), two programs employed during COVID-19 to mitigate some impacts of the COVID-19 disruption. While data are currently lacking, given the recency of the pandemic and the infrequency of income inequality measures, it appears as if government transfers (such as CERB and CESB) may have curtailed the rise of income inequality in the pandemic. Income inequality refers to the relative differences in incomes within a given group or area [7]. A wide gap in incomes is suggestive of a greater inequality in income, while a narrow gap suggests that there is not a wide distribution of incomes within a given group or area. In public health research, income inequality is most often measured using the Gini coefficient [8]. In Canada, the Gini coefficient has been increasing since the late 1980s and early 1990s [9, 10], especially in major urban centres [11]. For example, between 1990 and 2018, there was a 5.9% increase in the national Gini coefficient average in Canada [10].

## 2.0 Theory

The multitude of stressors associated with the COVID-19 pandemic may have led to diminished mental health among adolescents, and the effects are likely more pronounced in those

from socioeconomically disadvantaged areas. Increases in income inequality associated with the pandemic, including related increases in income inequality associated with job loss and loss of wages in the pandemic [6], may represent a key factor contributing to greater vulnerability to adverse mental health in adolescents during this period. This supposition is in line with current theories on the association between income inequality and mental health. For example, income inequality is associated with increased divestment in human capital (e.g., cuts to social spending), which has been common during the COVID-19 pandemic, with divestments in human capital defined as cuts or losses in, for example, training programs or mental healthcare (e.g., counselling services). In turn, such divestments are associated with worsened mental health, given human capital sectors are oftentimes those that can improve health [12]. That said, income inequality was likely to subsequently reduce following the onset of the CERB, which provided monthly payments to individuals who were affected by job or wage loss associated with COVID-19. CERB is one example of investment in human capital, which could reduce the effects of income inequality. Moreover, income inequality is associated with lowered social cohesion, which was exacerbated during the COVID-19 pandemic [3]. Among adolescents, social relationships and connections are especially important for mental well-being, considering that higher engagement and solidarity with peers can mitigate the harmful impacts of stress. With the onset of lockdowns and work-from-home orders, adolescents grew increasingly isolated and experienced increased stress during the pandemic [3]. This isolation, in turn, may have been associated with poorer mental health.

While existing research has linked income inequality in schools to adolescent depression [13], it is unclear if the pandemic exacerbated the association between income inequality and adolescent mental health. This work is important considering the significant toll the COVID-19 pandemic had on youth mental health. Furthermore, and to inform equitable distribution of supports and resources for pandemic recovery is essential and the prevention of sustained impacts. Also, the COVID-19 pandemic provides an important learning opportunity on how the relationship between income inequality and adolescent mental health may have been impacted in such circumstances, which may help to better prepare for and inform policies and programs in the case of future events. The objective of this study is to examine if the COVID-19 pandemic exacerbated the association between income inequality and adolescent depression and anxiety.

## 3.0 Material & methods

### 3.1 Data source: COMPASS, 2018–2021

The sample of this study included 29,722 adolescents (i.e., those in grades 9 through 12) in 129 schools and 43 Census divisions (CDs) in British Columbia, Alberta, and Ontario, and those in grades 7 to 11 in Quebec who participated in the Cannabis, Obesity, Mental health, Physical activity, Alcohol use, Smoking, and Sedentary behaviour (COMPASS) survey in the 2018/2019 academic year and responded at least once in the proceeding 2019/20 and 2020/21 survey waves. COMPASS is a prospective cohort study (2012–2027), which was designed to annually collect hierarchical longitudinal data from a convenience sample of secondary schools [14]. The in-class questionnaire uses active information passive consent protocols and collects student reported data related to behaviours, health, school connectedness, and academic outcomes [14]. Active information passive consent protocol means that information was distributed to parents regarding the survey and parents were required to contact the COMPASS team if they did not wish for their child/student to participate.

The current study used longitudinal data collected from 2018/19 to 2020/21. The data were deterministically linked to the 2016 Canadian Census to identify Census division (CD)-level

information. All procedures received ethics approval from the University of Waterloo (ORE#30118), Brock University (REB#18–099), University of Alberta (#RES0050375), CIUSSS de la Capitale-Nationale–Université Laval (#MP-13-2017-1264), and participating school boards.

## 3.2 Measures

**3.2.1 Exposure: Income inequality.**   The main exposure for this research was income inequality, measured using the Gini coefficient. The Gini coefficient is expressed a score between 0 and 1, with 1 indicating high inequality and 0 indicating perfect equality in incomes. The Gini coefficient was calculated using CanCHEC 2016 after-tax household income data in CDs. Gini coefficient calculations involve dividing the area between the Lorenz curve (i.e., the proportion of the total income of the population that is cumulatively earned by the lowest earners in the population) of an income distribution and the distribution line of incomes within an area by the area under the distribution line of the Lorenz curve [15]. Gini coefficient was z-transformed to improve interpretability.

**3.2.2 Outcome measures: Adolescent depressive and anxiety symptoms.**   Depressive and anxiety symptoms were assessed using self-report measures that have demonstrated validity in adolescents [16–18], and measurement invariance by gender specifically in the COMPASS survey [19]. The 10-item Center for Epidemiologic Studies Depression scale Revised (CES-D) asked students how often they experience symptoms within the last 7 days [20] and the 7-item Generalized Anxiety Disorder scale (GAD-7) was used to capture anxiety scores by asking how often students experienced each symptom in the past two weeks [21]. For example, the item "Please indicate how often the following statements apply to you: I felt depressed", was used to calculate a continuous depression score; whereas an item such as "Over the last 2 weeks, how often have you been bothered by the following problems? Feeling nervous, anxious, or on edge", was used to calculate a continuous anxiety score. Measures of depression and anxiety were standardized using the z-transformation.

**3.2.3 Time: COVID-19.**   Those observations recorded prior to the March 2020 school closures associated with COVID-19 were coded as "pre-COVID-19" [time(T)1], and those following school closures and in the 2020/2021 wave were coded as "peri-COVID-19" (i.e., "during" COVID-19; T2). For the 2019–20 survey wave, students that responded to the questionnaire prior to school closures due to COVID-19 were coded as "pre-COVID-19" responses, whereas students answered following the closures were regarded as "peri-COVID-19".

**3.2.4 Covariates.**   *3.2.4.1 Individual-level measures.* The models adjusted for age (in years), gender (male, female, prefer not to say, other), race/ethnicity [Black, Hispanic, Asian, or other (selected 'other', multiple responses, or Métis, First Nations, or Inuit as ethics restrictions precluded the identification of students with Indigenous heritage for separate study [22])], and personal spending money (in increments, as available in the COMPASS) at baseline. No measures for individual-level income exist in COMPASS; therefore, personal spending money of students served as a proxy.

*3.2.4.2 School-level measures.* The analyses adjusted for whether the school was categorized as a "private" or "public" institution at baseline.

*3.2.4.3 Census division-level measures.* Characteristics of CDs across British Columbia, Alberta, Ontario, and Quebec were used. CD are provincially legislated areas or their equivalents, such as regional districts or counties. Models adjusted for median CD-level after-tax household income, proportion of immigrant (in the past five years) households, proportion of visible minority (i.e., not white, or Indigenous) households, and the proportion of lone-parent (i.e., single-parent) households in each CD.

### 3.3 Statistical analysis: Three-level multi-level modelling

Three-level multi-level models were used to investigate the association between CD-level income inequality and each mental health outcome (depression score, anxiety score) over time (e.g., pre- versus peri-COVID-19; level-1) while controlling for both individual (level-2) and CD-level (level-3) characteristics [23]. Multi-level models were necessary because repeated measures were clustered within students, who were clustered within CDs. Models were developed using a step-up method. First, intercept-only models were fit to determine the Intraclass Correlation Coefficient (ICC) and quantify the proportion of variance in depressive and anxiety scores explained at each level in each model. Next, bivariate, multi-level analyses helped to identify the unadjusted association between CD-level income inequality and each of the outcomes over time. After this, adjusted models were fit that included both CD-level, school-level, and individual-level factors, as well as a cross-level interaction term between CD-level income inequality and time (e.g., peri-COVID-19), to examine if COVID-19 changed the association between income inequality and adolescent mental health. Case-complete data was used.

**3.3.1 Weighting.** There was an increase in non-response in the spring of 2020, due to COVID-19 because school closures limited the ability for administrators to distribute surveys. As a result of school closures, COMPASS was administered online, and the response rates were reduced from 83% to 58% [Personal communication with Angelica Amores, COMPASS on December 6, 2021]. The 2019/20 and 2020/21 survey waves were weighted using sampling weights to account for non-response owing to COVID-19.

## 4.0 Results

The study sample included 29,722 students across 43 Census divisions in British Columbia, Alberta, Ontario, and Quebec. The average age of the sample at baseline was 14.9 years [standard deviation (SD) = 1.5] and ranged between 12 and 19 years of age. The majority of the sample self-reported as white (76.3%), female (54.4%), and had weekly spending money of over $100 (20.9%). The mean Gini coefficient at the CD-level was 0.37 (SD = 0.03) and the median after-tax household income was $58,891.70 (SD = $8,843.50). The average proportion of visible minority households, single-parent households, and recent immigrant households were 9.9%, 15.4%, and 1.6%, respectively, across CDs. Details on the distribution of the sample are available in Table 1. Null models indicated an ICC for depressive scores of 0.02 (95% CI = 0.01, 0.04) within CDs and 0.27 (95% CI = 0.26, 0.28) for students over time within CDs. The ICCs for the anxiety scores were 0.03 (95% CI = 0.02, 0.05) within CDs and 0.32 (0.32, 0.34) for students over time within CDs.

In an unadjusted analysis, the results demonstrated that a one SD-unit increase in Gini coefficient was associated with a 0.08 score increase in depressive scores (95% CI = 0.04, 0.11). Students who completed the COMPASS survey after the onset of COVID reported 0.20-unit higher depressive scores (95% CI = 0.16, 0.24) compared to their responses pre-COVID. However, the association between income inequality and depressive scores did not change in magnitude by the onset of COVID-19. Findings were similar in the adjusted analyses, with students who completed their COMPASS questionnaire following the onset of COVID-19 reporting higher depressive scores (B = 0.14, 95% CI = 0.10, 0.18) compared to pre-COVID. However, the association between income inequality and depressive scores was eliminated, and no interaction observed between z-transformed Gini coefficient and COVID-19 onset when adjusting for relevant covariates. More details are available in Table 2.

For anxiety scores, the unadjusted model demonstrated an association between income inequality and anxiety scores (B = 0.09, 95% CI = 0.05, 0.13), COVID onset and anxiety scores (B = 0.19, 95% CI = 0.17, 0.21), and the interaction between z-transformed income inequality

**Table 1. Table describing the distribution of the COMPASS sample in 2018/19.**

| | | *n* | % |
|---|---|---|---|
| **Individual-level characteristics** | | | |
| Gender | Girl | 16,077 | 54.1 |
| | Boy | 13,479 | 45.3 |
| | Prefer not to say | 166 | 0.01 |
| Race/Ethnicity | White | 22,517 | 76.25 |
| | Black | 929 | 3.15 |
| | Asian | 2,426 | 8.22 |
| | Hispanic | 838 | 2.84 |
| | Other | 2,820 | 9.55 |
| Weekly Spending Money | $0 | 4,856 | 20.45 |
| | $1 to $5 | 1,869 | 7.87 |
| | $6 to $10 | 2,132 | 8.98 |
| | $11 to $20 | 3,604 | 15.18 |
| | $21 to $40 | 3,070 | 12.93 |
| | $41 to $100 | 3,256 | 13.71 |
| | More than $100 | 4,958 | 20.88 |
| | | **Mean** | **SD** |
| | Age | 14.88 | 1.52 |
| **School-level characteristics** | | ***n*** | % |
| | Private | 8 | 6.20 |
| | Public | 121 | 93.80 |
| **CD-level characteristics** | | | |
| | Gini coefficient | 0.37 | 0.03 |
| | Visible minority % | 0.10 | 0.12 |
| | Lone parent % | 0.15 | 0.02 |
| | Recent immigrant % | 0.02 | 0.01 |
| | | **Median** | **SD** |
| | After-tax household income | 58891.70 | 8843.50 |

and COVID-19 onset and anxiety scores (B = 0.02, 95% CI = 0.001, 0.04). The adjusted analyses indicated that the association between income inequality on anxiety scores was changed following the onset of COVID-19 (B = 0.02, 95% CI = 0.0004, 0.03). More details are available in Table 3. A graphical representation of the interaction between z-transformed Gini coefficient and COVID-19 onset on anxiety scores is available in Fig 1. Fig 1 depicts the estimated probabilities of anxiety as associated with CD-level income inequality over time. This figure demonstrates how the gaps in anxiety scores between higher and lower income inequality groups widened after the onset of the pandemic. Further, the largest increases in anxiety scores were amongst those with the highest levels of income inequality at baseline and estimated probabilities of anxiety increased after COVID-19 regardless of income inequality level.

## 5.0 Discussion

The results of this study indicated that the association between income inequality and anxiety was exacerbated by the onset of COVID-19, meaning that adolescents attending schools in higher income inequality areas had larger increases in anxiety symptoms from before to during the pandemic than those attending schools in lower income inequality areas. This study also contributes to the existing literature that reports an association between higher income

**Table 2. Table describing the unadjusted and adjusted association between income inequality and depressive scores in COVID-19, COMPASS 2018–2021.**

| | | Unadjusted model | | | | Adjusted model | | | |
|---|---|---|---|---|---|---|---|---|---|
| | | 95% CI | | | | 95% CI | | | |
| | | Coef. | Lower bound | Upper bound | p-value | Coef. | Lower bound | Upper bound | p-value |
| CD-level z-transformed Gini coefficient (zGini) | | 0.08 | 0.04 | 0.11 | <0.001 | 0.02 | -0.005 | 0.04 | 0.124 |
| COVID | (ref: pre-COVID) | 0.20 | 0.16 | 0.24 | <0.001 | 0.14 | 0.10 | 0.18 | <0.001 |
| Interaction term | zGini * COVID | 0.01 | -0.02 | 0.03 | 0.660 | 0.0002 | -0.02 | 0.03 | 0.98 |
| % of visible minority households | | | | | | -0.02 | -0.07 | 0.02 | 0.331 |
| % of lone-parent households | | | | | | 0.04 | 0.02 | 0.07 | <0.001 |
| % of recent immigrant households | | | | | | 0.01 | -0.04 | 0.04 | 0.964 |
| Median household income | | | | | | 0.07 | 0.05 | 0.09 | <0.001 |
| Gender | Boy (ref: Girl) | | | | | -0.50 | -0.53 | -0.47 | <0.001 |
| | Other description | | | | | 0.71 | 0.61 | 0.82 | <0.001 |
| | Prefer not to say | | | | | 0.30 | 0.21 | 0.38 | <0.001 |
| Race/Ethnicity | Black (ref: non-white) | | | | | 0.04 | -0.002 | 0.08 | 0.064 |
| | Asian | | | | | 0.05 | 0.02 | 0.08 | 0.001 |
| | Hispanic | | | | | 0.09 | 0.06 | 0.13 | <0.001 |
| | Other | | | | | 0.19 | 0.16 | 0.23 | <0.001 |
| Weekly spending money | $1 to $5 (ref: $0) | | | | | -0.04 | -0.06 | -0.01 | <0.001 |
| | $6 to $10 | | | | | -0.07 | -0.10 | -0.04 | 0.003 |
| | $11 to $20 | | | | | -0.08 | -0.12 | -0.05 | <0.001 |
| | $21 to $40 | | | | | -0.09 | -0.13 | -0.06 | <0.001 |
| | $41 to $100 | | | | | -0.10 | -0.15 | -0.05 | <0.001 |
| | More than $100 | | | | | -0.07 | -0.10 | -0.03 | 0.001 |
| Age | In years | | | | | 0.05 | 0.05 | 0.06 | <0.001 |
| School type | Private (ref: Public) | | | | | -0.10 | -0.12 | -0.08 | <0.001 |
| Intercept | | -0.03 | -0.07 | 0.02 | 0.239 | -0.55 | -0.67 | -0.43 | <0.001 |

inequality and increased depression and anxiety in adolescents [13, 24]. The results also indicated that income inequality was not associated with depressive score following the onset of COVID-19. Unsurprisingly, the results also indicated that following the onset of COVID-19, both anxiety and depressive scores increased in the sample, as our sample aged throughout the study period.

While income inequality and rates of adolescent adverse mental health are likely on the rise since COVID-19, the association between income inequality and mental health peri-pandemic had not yet been studied. This work is the first to quantify the joint effects of the COVID-19 pandemic and income inequality on adolescent mental health. While earlier work has examined the association between income inequality and adolescent mental health, most studies were cross-sectional in nature; as such, it is unclear if income inequality preceded mental health problems in these studies and the quality of existing evidence is quite low. This work shows that income inequality at baseline is associated with increases in depressive and anxiety symptoms over time.

While no association between income inequality and depression was observed in this study, the direction of effect is still what we might expect resulting from the COVID-19 pandemic, with higher income inequality being associated with greater increases in depressive scores. It is possible that these findings are attributable to the complex mechanisms that underlie mental health. For example, researchers have demonstrated that feelings of anxiety oftentimes present

**Table 3. Table describing the unadjusted and adjusted association between income inequality and anxiety scores in COVID-19, COMPASS 2018–2021.**

| | | Unadjusted model | | | | Adjusted model | | | |
|---|---|---|---|---|---|---|---|---|---|
| | | 95% CI | | | | 95% CI | | | |
| | | Coef. | Lower bound | Upper bound | p-value | Coef. | Lower bound | Upper bound | p-value |
| CD-level z-transformed Gini coefficient (zGini) | | 0.09 | 0.05 | 0.13 | <0.001 | 0.02 | -0.01 | 0.05 | 0.173 |
| COVID | (ref: pre-COVID) | 0.19 | 0.17 | 0.21 | <0.001 | 0.13 | 0.11 | 0.15 | <0.001 |
| Interaction term | zGini * COVID | **0.02** | **0.00** | **0.04** | **0.038** | **0.02** | **0.01** | **0.03** | **0.04** |
| % of visible minority households | | | | | | -0.02 | -0.08 | 0.04 | 0.431 |
| % of lone-parent households | | | | | | 0.07 | 0.05 | 0.10 | <0.001 |
| % of recent immigrant households | | | | | | 0.003 | -0.06 | 0.05 | 0.992 |
| Median household income | | | | | | 0.08 | 0.06 | 0.11 | <0.001 |
| Gender | Boy (ref: Girl) | | | | | -0.60 | -0.64 | -0.55 | <0.001 |
| | Other description | | | | | 0.49 | 0.37 | 0.61 | <0.001 |
| | Prefer not to say | | | | | 0.09 | -0.02 | 0.20 | 0.102 |
| Race/Ethnicity | Black (ref: non-white) | | | | | -0.11 | -0.16 | -0.06 | <0.001 |
| | Asian | | | | | -0.09 | -0.12 | -0.06 | <0.001 |
| | Hispanic | | | | | 0.03 | -0.03 | 0.04 | 0.844 |
| | Other | | | | | 0.12 | 0.09 | 0.15 | <0.001 |
| Weekly spending money | $1 to $5 (ref: $0) | | | | | -0.02 | -0.05 | 0.01 | 0.123 |
| | $6 to $10 | | | | | -0.05 | -0.08 | -0.02 | 0.001 |
| | $11 to $20 | | | | | -0.07 | -0.10 | -0.04 | <0.001 |
| | $21 to $40 | | | | | -0.06 | -0.09 | -0.03 | <0.001 |
| | $41 to $100 | | | | | -0.05 | -0.09 | -0.01 | 0.007 |
| | More than $100 | | | | | -0.01 | -0.04 | 0.03 | 0.774 |
| Age | In years | | | | | 0.05 | 0.04 | 0.06 | <0.001 |
| School type | Private (ref: Public) | | | | | -0.08 | -0.09 | -0.06 | <0.001 |
| Intercept | | -0.01 | -0.07 | 0.04 | 0.580 | -0.43 | -0.54 | -0.33 | <0.001 |

before those of depressive symptoms and these conditions are highly comorbid [25–27] and anxiety is more likely to emerge in pre- and mid-adolescence, whereas depression is more commonly onset during adolescence or in early adulthood [26]. Moreover, it is possible that the lack of statistical significance is owing a lag period, by which income inequality may take several years before the impacts incur. This, in tandem with the previous point, may explain the null association between income inequality and depression during the COVID-19 pandemic. Future longitudinal research is needed to examine potential delayed and/or sustained impacts of income inequality into the ongoing and recovery phases of the pandemic.

Mental health in adolescents has likely worsened since the onset of the COVID-19 pandemic, perhaps owing to worry regarding the virus, but also to isolation and social segregation resulting from public health measures (e.g., school closures, food insecurity, worries about the future). That said, results from a multi-national study indicated that countries that enacted stringent public health orders sooner actually had lower rates of depression in comparison to those that did not implement such measures with haste [28]. While some studies have demonstrated that mental health in children did not, in fact, worsen as a result of the pandemic [29], albeit this study was conducted amongst elementary-school-aged children (ages 9 to 12) who are at lower risk for mental health conditions given that most mental health conditions are onset in adolescence and young adulthood.

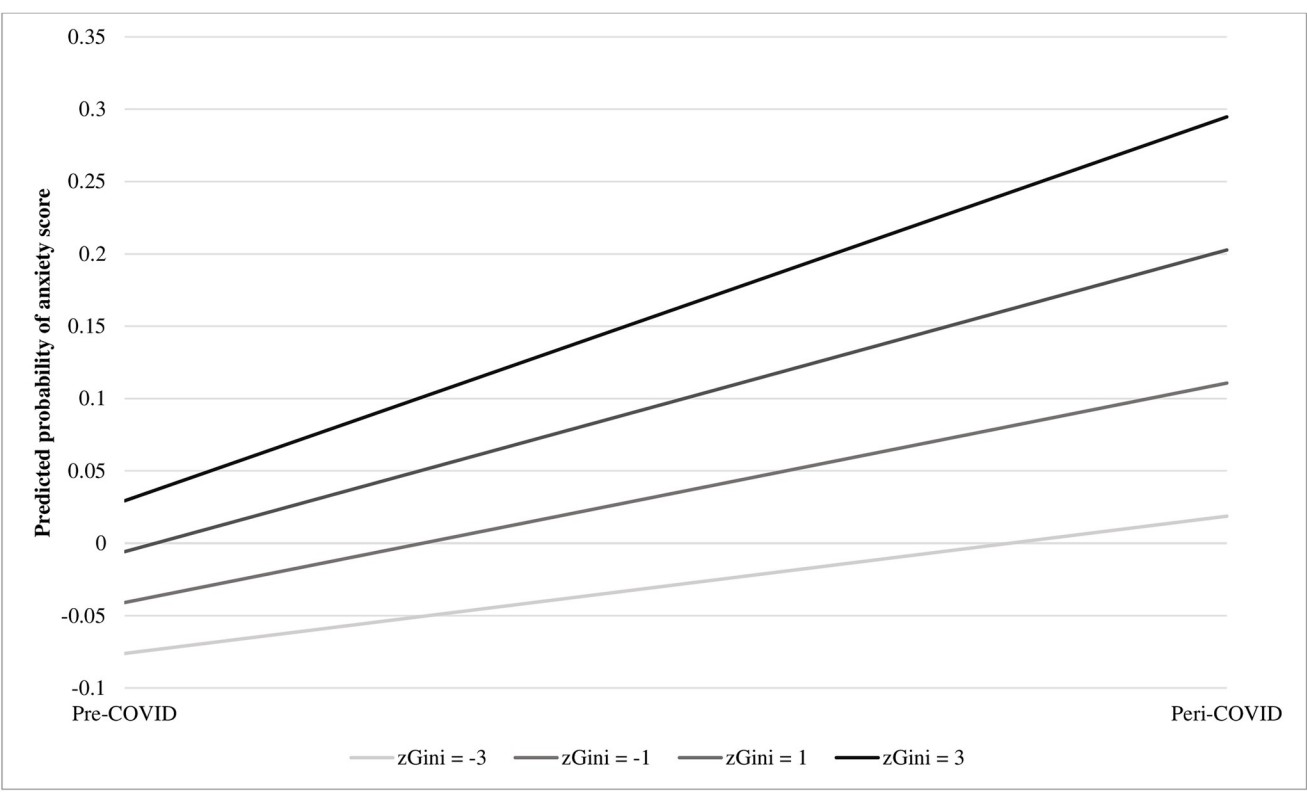

**Fig 1. Line graph demonstrating the graphical representation of the interaction between z-transformed Gini coefficient (at values of -3, -1, 1, and 3) and COVID-19 onset (pre, versus peri-) on the predicted probability of anxiety scores, COMPASS 2018/19-2020/21.**

The current study used a large, school-based dataset to investigate the research question and validated measures of depression and anxiety (i.e., the CES-D and GAD-7, respectively) and included secondary school students across four Canadian provinces. One survey wave (2019/20) had a reduced response rate owing to the onset of the pandemic and shift to online learning and COMPASS distribution, so study weights were developed and applied to improve the generalizability to the typical COMPASS sample. Moreover, this study is the first to investigate the association between income inequality and adolescent mental health during COVID-19 and lends to a growing body of evidence that suggests that income inequality is harmful for adolescent mental health.

Existing research on mental health during COVID-19 is limited in that many report on mental health by asking individuals to compare how they are currently feeling peri-pandemic compared to pre-pandemic [30–33]. Measures of pre-pandemic mental health may be biased by individuals current emotional state when responding peri-pandemic. The current study addresses this limitation by using measures of mental health taken before the pandemic, and once again during the pandemic.

## 5.1 Limitations

The results of the current study should be interpreted bearing in mind limitations. For example, it is important to note that while COMPASS draws upon many schools across BC, Alberta, Ontario, and Quebec, convenience sampling techniques are used; therefore, the findings of this work may have limited generalizability to larger populations. However, it is likely that the

results could be applied to schools with similar attributes. Recall bias is also possible but likely minimized in the current study because youth were asked to report on feelings of depression in the past week and anxiety in the past-two weeks. Past-week and past-two-week behaviours are likely easier to recall than longer time periods. Reporting bias could occur in terms of responding to the COMPASS survey, in that social desirability bias may occur when youth self-report on their symptoms of mental health and other factors considered in our modelling. Moreover, responding to questionnaires online as a result of school closures could bias the results; that said, by controlling for pre- versus peri-pandemic responses we likely alleviate some of this bias.

Another important limitation of the proposed study is residual confounding. Additional confounders and mediators might exist that explain some of the variability in the association between income inequality and adolescent depression and anxiety. For example, lower earners have increased risk of mood disorders compared to higher earners [34, 35]; as such, it is important to account for absolute income in studies of income inequality to identify independent effects of income inequality on outcomes [36]. The COMPASS survey does not have information on students' household income, and so we were unable to control for absolute income at the individual-level. Considering this limitation, the median household income at the CD-level and weekly spending money at the individual-level were included in modeling to account for absolute income.

Further research is necessary to improve our understanding of the association between income inequality and adolescent mental health during COVID-19. For example, quasi-experimental designs could be employed to test the consequences of COVID-19 on the association between income inequality and adolescent mental health.

## 6.0 Conclusions

The current study demonstrated that income inequality is more highly associated with increased anxiety symptoms during COVID-19 than prior to COVID-19. These results are directly applicable to both school-level and upstream policies for mental health and demonstrate the importance income inequality. This study also contributes to the literature documenting the potential harms associated with COVID-19 and income inequality.

## Author Contributions

**Conceptualization:** Paul J. Veugelers, Roman Pabayo.

**Data curation:** Karen A. Patte, Scott T. Leatherdale.

**Formal analysis:** Claire Benny, Roman Pabayo.

**Funding acquisition:** Karen A. Patte, Brendan T. Smith, Scott T. Leatherdale.

**Methodology:** Claire Benny.

**Supervision:** Ambikaipakan Senthilselvan, Paul J. Veugelers, Roman Pabayo.

**Writing – original draft:** Claire Benny.

**Writing – review & editing:** Claire Benny, Ambikaipakan Senthilselvan, Karen A. Patte, Brendan T. Smith, Paul J. Veugelers, Scott T. Leatherdale, Roman Pabayo.

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
