## [Decision Letter · Decision Letter 0]

4 Jul 2023

PONE-D-23-14182Income inequality and mental health in adolescents during COVID-19, results from COMPASS 2018-2021.PLOS ONE

Dear Dr. Benny,

Thank you for submitting your manuscript to PLOS ONE. After careful consideration, we feel that it has merit but does not fully meet PLOS ONE’s publication criteria as it currently stands. Therefore, we invite you to submit a revised version of the manuscript that addresses the points raised during the review process.

We look forward to receiving your revised manuscript.

Kind regards,

Md. Saiful Islam, BPH, MPH

Academic Editor

PLOS ONE

Journal Requirements:

4. Please include your tables as part of your main manuscript and remove the individual files. Please note that supplementary tables (should remain/ be uploaded) as separate "supporting information" files

Reviewers' comments:

Reviewer's Responses to Questions

**Comments to the Author**

1. Is the manuscript technically sound, and do the data support the conclusions?

Reviewer #1: Yes

Reviewer #2: Yes

2. Has the statistical analysis been performed appropriately and rigorously? 

Reviewer #1: Yes

Reviewer #2: Yes

3. Have the authors made all data underlying the findings in their manuscript fully available?

Reviewer #1: Yes

Reviewer #2: No

4. Is the manuscript presented in an intelligible fashion and written in standard English?

Reviewer #1: Yes

Reviewer #2: Yes

5. Review Comments to the Author

Reviewer #1: The authors Claire Benny et al “ Income inequality and mental health in adolescents during COVID-19, results from COMPASS 2018-2021” provides income inequality associated with higher anxiety scores, but the association with depressive scores is mediated by other covariates.

Major Comments:

1)The author need to provide more evidence of how income inequality has been affected by the COVID-19 pandemic in Canada and its potential implications for mental health outcomes.

2)It would be useful to provide a brief explanation of what is meant by "divestment in human capital" for readers who may not be familiar with the term.

3) It would be helpful to elaborate on how social cohesion may influence adolescent mental health outcomes and provide supporting evidence or theories that explain this relationship.

4)It would be beneficial to include a brief statement or rationale for why understanding this association is important or relevant in the context of public health or mental health research.

5)It would be helpful to provide a brief background on the significance of neuroendocrine transformation in lung cancer and its impact on treatment outcomes.

6)The introduction could benefit from a more detailed explanation of the clinical implications and importance of understanding this molecular heterogeneity.

7)The author need to proofread the manuscript for any grammatical errors.

Reviewer #2: researchers should carefully select a single appropriate venue for their work and adhere to the publication policies and guidelines of that journal or conference. It is crucial to prioritize transparency, integrity, and responsible conduct when disseminating research findings.Many journals require authors to transfer their copyright or grant exclusive publication rights. Dual publication violates these agreements, as it involves submitting the same work to multiple venues without proper authorization.

6. PLOS authors have the option to publish the peer review history of their article (what does this mean?). If published, this will include your full peer review and any attached files.

Reviewer #1: **Yes: **Naresh Poondla

Reviewer #2: **Yes: **Nigar Arif Poladlı

---

## [Author Response · Author response to Decision Letter 0]

12 Jul 2023

Response to Reviewers

Reviewer #1: 

The authors Claire Benny et al “ Income inequality and mental health in adolescents during COVID-19, results from COMPASS 2018-2021” provides income inequality associated with higher anxiety scores, but the association with depressive scores is mediated by other covariates.

Response: Thank you for reviewing our manuscript. 

Major Comments:

1)The author need to provide more evidence of how income inequality has been affected by the COVID-19 pandemic in Canada and its potential implications for mental health outcomes.

Response: Thank you for your comment. We have commented further on speculated changes to income inequality in COVID-19. See page 5, paragraph 2 for this addition.

Data on the effects of the COVID-19 pandemic on income inequality are currently lacking, given the recency of the pandemic and the infrequency of income inequality measures. It is plausible that the pandemic widened the gap between the highest and lowest earners (i.e., income inequality) in Canada, with increases in unemployment rates disproportionately affecting the lowest earners, those less educated, immigrants, people of colour, and those with precarious employment (Statistics Canada, 2020a); that said, income inequality may have also been subsequently decreasing owing to the Canadian Emergency Response Benefit (CERB) and the Canadian Emergency Student Benefit (CESB), two programs employed during COVID-19 to mitigate some impacts of the COVID-19 disruption. While data are currently lacking, given the recency of the pandemic and the infrequency of income inequality measures, it appears as if government transfers (such as CERB and CESB) may have curtailed the rise of income inequality in the pandemic. Income inequality refers to the relative differences in incomes within a given group or area (Left Business Observer, 1993). A wide gap in incomes is suggestive of a greater inequality in income, while a narrow gap suggests that there is not a wide distribution of incomes within a given group or area. In public health research, income inequality is most often measured using the Gini coefficient (De Maio, 2007). In Canada, the Gini coefficient has been increasing since the late 1980s and early 1990s (Breau, 2014; Statistics Canada, 2020b), especially in major urban centres (Abedi, 2017). For example, between 1990 and 2018, there was a 5.9% increase in the national Gini coefficient average in Canada (Statistics Canada, 2020b). 

2)It would be useful to provide a brief explanation of what is meant by "divestment in human capital" for readers who may not be familiar with the term.

Response: Thank you for this comment. We believe providing a definition of “divestment in human capital” would strengthen and improve readability of the manuscript. Please see page 6, paragraph 1 for an explanation of this term.

The multitude of stressors associated with the COVID-19 pandemic may have led to diminished mental health among adolescents, and the effects are likely more pronounced in those from socioeconomically disadvantaged areas. Increases in income inequality associated with the pandemic, including related increases in income inequality associated with job loss and loss of wages in the pandemic (Statistics Canada, 2020a), may represent a key factor contributing to greater vulnerability to adverse mental health in adolescents during this period. This supposition is in line with current theories on the association between income inequality and mental health. For example, income inequality is associated with increased divestment in human capital (e.g., cuts to social spending), which has been common during the COVID-19 pandemic, with divestments in human capital defined as cuts or losses in, for example, training programs or mental healthcare (e.g., counselling services). In turn, such divestments are associated with worsened mental health, given human capital sectors are oftentimes those that can improve health (Scheffler et al., 2010). That said, income inequality was likely to subsequently reduce following the onset of the CERB, which provided $2,000 monthly payments to individuals who were affected by job or wage loss association with COVID-19. CERB is one example of investment in human capital, which could reduce the effects of income inequality. Moreover, income inequality is associated with lowered social cohesion, which was exacerbated during the COVID-19 pandemic (Stephenson, 2021). Among adolescents, social relationships and connections are especially important for mental well-being, considering that higher engagement and solidarity with peers can mitigate the harmful impacts of stress. With the onset of lockdowns and work-from-home orders, adolescents grew increasingly isolated and experienced increased stress during the pandemic. This isolation, in turn, may have been associated with poorer mental health. 

3) It would be helpful to elaborate on how social cohesion may influence adolescent mental health outcomes and provide supporting evidence or theories that explain this relationship.

Response: We have included further evidence of the link between social cohesion and adolescent mental health outcomes in the revised manuscript. See page 6, paragraph 1.

The multitude of stressors associated with the COVID-19 pandemic may have led to diminished mental health among adolescents, and the effects are likely more pronounced in those from socioeconomically disadvantaged areas. Increases in income inequality associated with the pandemic, including related increases in income inequality associated with job loss and loss of wages in the pandemic (Statistics Canada, 2020a), may represent a key factor contributing to greater vulnerability to adverse mental health in adolescents during this period. This supposition is in line with current theories on the association between income inequality and mental health. For example, income inequality is associated with increased divestment in human capital (e.g., cuts to social spending), which has been common in COVID-19, with divestments in human capital being defined as cuts or losses in, for example, training programs or healthcare. In turn, such divestments are associated with worsened mental health, given human capital sectors are oftentimes those that can improve health (Scheffler et al., 2010). That said, income inequality was likely to subsequently reduce following the onset of the CERB, which provided monthly payments to individuals who were affected by job or wage loss associated with COVID-19. CERB is one example of investment in human capital, which could reduce the effects of income inequality. Moreover, income inequality is associated with lowered social cohesion, which was exacerbated during the COVID-19 pandemic (Stephenson, 2021). Among adolescents, social relationships and connections are especially important for mental well-being, considering that higher engagement and solidarity with peers can mitigate the harmful impacts of stress. With the onset of lockdowns and work-from-home orders, adolescents grew increasingly isolated and experienced increased stress during the pandemic (Stephenson, 2021). This isolation, in turn, may have been associated with poorer mental health. 

4)It would be beneficial to include a brief statement or rationale for why understanding this association is important or relevant in the context of public health or mental health research.

Response: We have provided a brief statement in the concluding paragraph of the introduction to support why this research is important in the context of public health, please see page 7, paragraph 1.

While existing research has linked income inequality in schools to adolescent depression (Benny et al., 2022), it is unclear if the pandemic exacerbated the association between income inequality and adolescent mental health. This work is important to inform equitable distribution of supports and resources for pandemic recovery and the prevention of sustained impacts. Also, the COVID-19 pandemic provides an important learning opportunity on how the relationship between income inequality and adolescent mental health may have been impacted in such circumstances, which may help to better prepare for and inform policies and programs in the case of future events. The objective of this study is to examine if the COVID-19 pandemic exacerbated the association between income inequality and adolescent depression and anxiety. 

5)It would be helpful to provide a brief background on the significance of neuroendocrine transformation in lung cancer and its impact on treatment outcomes.

Response: Thank you for your comment. Given that the focus of this paper is on income inequality and adolescent mental health in COVID-19, the authors and myself feel that explanations on neuroendocrine transformation in lung cancer is out of scope in this manuscript. 

6)The introduction could benefit from a more detailed explanation of the clinical implications and importance of understanding this molecular heterogeneity.

Response: Given that this paper is grounded within population health perspectives and the word limits of the journal, we believe it is out of scope for the authors to comment on the molecular heterogeneity mentioned by Reviewer 1. 

7)The author need to proofread the manuscript for any grammatical errors.

Response: We have read over the text and revised some grammatical errors. Thank you for this comment. See, for example, page 10, paragraph 1:

The models adjusted for age (in years), gender (male, female, prefer not to say, other), race/ethnicity [Black, Hispanic, Asian, or other (selected ‘other’, multiple responses, or Métis, First Nations, or Inuit as ethics restrictions precluded the identification of students with Indigenous heritage for separate study (Research, n.d.))], and personal spending money (in increments, as available in the COMPASS) at baseline. No measures for individual-level income exist in COMPASS; therefore, personal spending money of students served as a proxy.

Reviewer #2: 

[R]esearchers should carefully select a single appropriate venue for their work and adhere to the publication policies and guidelines of that journal or conference. It is crucial to prioritize transparency, integrity, and responsible conduct when disseminating research findings. Many journals require authors to transfer their copyright or grant exclusive publication rights. Dual publication violates these agreements, as it involves submitting the same work to multiple venues without proper authorization.

Response: Thank you for voicing your concerns re: dual publication. This manuscript has only been submitted to PLOS One, in line with journal standards. Please let us know if there are further concerns with this work and we would be more than happy to accommodate.

---

## [Decision Letter · Decision Letter 1]

2 Aug 2023

PONE-D-23-14182R1Income inequality and mental health in adolescents during COVID-19, results from COMPASS 2018-2021.PLOS ONE

Dear Dr. Benny,

Thank you for submitting your manuscript to PLOS ONE. After careful consideration, we feel that it has merit but does not fully meet PLOS ONE’s publication criteria as it currently stands. Therefore, we invite you to submit a revised version of the manuscript that addresses the points raised during the review process. Please submit your revised manuscript by Sep 16 2023 11:59PM. If you will need more time than this to complete your revisions, please reply to this message or contact the journal office at plosone@plos.org. Please include the following items when submitting your revised manuscript:A rebuttal letter that responds to each point raised by the academic editor and reviewer(s). You should upload this letter as a separate file labeled 'Response to Reviewers'.A marked-up copy of your manuscript that highlights changes made to the original version. You should upload this as a separate file labeled 'Revised Manuscript with Track Changes'.An unmarked version of your revised paper without tracked changes. You should upload this as a separate file labeled 'Manuscript'.We look forward to receiving your revised manuscript.

Kind regards,

Md. Saiful Islam, BPH, MPH

Academic Editor

PLOS ONE

Journal Requirements:

Additional Editor Comments:

Please address the reviewer 2 comments.

Reviewers' comments:

Reviewer's Responses to Questions

**Comments to the Author**

1. If the authors have adequately addressed your comments raised in a previous round of review and you feel that this manuscript is now acceptable for publication, you may indicate that here to bypass the “Comments to the Author” section, enter your conflict of interest statement in the “Confidential to Editor” section, and submit your "Accept" recommendation.

Reviewer #1: All comments have been addressed

Reviewer #2: All comments have been addressed

2. Is the manuscript technically sound, and do the data support the conclusions?

Reviewer #1: Yes

Reviewer #2: Yes

3. Has the statistical analysis been performed appropriately and rigorously? 

Reviewer #1: Yes

Reviewer #2: No

4. Have the authors made all data underlying the findings in their manuscript fully available?

Reviewer #1: Yes

Reviewer #2: No

5. Is the manuscript presented in an intelligible fashion and written in standard English?

Reviewer #1: Yes

Reviewer #2: Yes

6. Review Comments to the Author

Reviewer #1: (No Response)

Reviewer #2: This research shows promise. Moreover, I commend the author for addressing research ethics and encourage adherence to publication ethics guidelines throughout the submission process. Keep up the good work and ensure a transparent and robust research process to contribute meaningfully to your field.

7. PLOS authors have the option to publish the peer review history of their article (what does this mean?). If published, this will include your full peer review and any attached files.

Reviewer #1: **Yes: **Naresh Poondla

Reviewer #2: **Yes: **Nigar Arif-Poladlı

---

## [Author Response · Author response to Decision Letter 1]

15 Sep 2023

Response to Editor

Editor: 

Response: We have revisited the reference list for our manuscript and confirm that, to our knowledge, no papers cited have been retracted. Thank you for your comment.

Response to Reviewers

Reviewer #2: 

[R]esearchers should carefully select a single appropriate venue for their work and adhere to the publication policies and guidelines of that journal or conference. It is crucial to prioritize transparency, integrity, and responsible conduct when disseminating research findings. Many journals require authors to transfer their copyright or grant exclusive publication rights. Dual publication violates these agreements, as it involves submitting the same work to multiple venues without proper authorization.

Response: Thank you for voicing your concerns re: dual publication. This manuscript has only been submitted to PLOS One, in line with journal standards. In terms of transparency, we have amended our data availability statement to align with the journal standards. Please note that the data used for our study cannot be included with our submission because of concerns with anonymity of the respondents. We have provided contact information for COMPASS data access, should this reviewer or any reader be interested in garnering access to the data used in this study. Please see page 2, paragraph 2 in the revised text.

Data availability. Data cannot be shared publicly because of ethical and confidentiality concerns. Data are available from the COMPASS team at the University of Waterloo (contact via sherry.rezvani@uwaterloo.ca) for researchers who meet the criteria for access to confidential data.

---

## [Editor Report · Decision Letter 2]

8 Oct 2023

Income inequality and mental health in adolescents during COVID-19, results from COMPASS 2018-2021.

PONE-D-23-14182R2

Dear Dr. Claire Benny,

We’re pleased to inform you that your manuscript has been judged scientifically suitable for publication and will be formally accepted for publication once it meets all outstanding technical requirements.

Kind regards,

Md. Saiful Islam, BPH, MPH

Academic Editor

PLOS ONE

---

## [Editor Report · Acceptance letter]

16 Oct 2023

PONE-D-23-14182R2 

Income inequality and mental health in adolescents during COVID-19, results from COMPASS 2018-2021. 

Dear Dr. Benny:

I'm pleased to inform you that your manuscript has been deemed suitable for publication in PLOS ONE. Congratulations! Your manuscript is now with our production department. 

Kind regards, 

on behalf of

Dr. Md. Saiful Islam 

Academic Editor

PLOS ONE